# There and Back Again: Hox Clusters Use Both DNA Strands

**DOI:** 10.3390/jdb9030028

**Published:** 2021-07-15

**Authors:** Elena L. Novikova, Milana A. Kulakova

**Affiliations:** 1Department of Embryology, St. Petersburg State University, Universitetskaya nab. 7–9, 199034 Saint Petersburg, Russia; e.nowikowa@spbu.ru; 2Laboratory of Evolutionary Morphology, Zoological Institute RAS, Universitetskaya nab. 1, 199034 Saint Petersburg, Russia

**Keywords:** long noncoding RNAs, lncRNAs, antisense ncRNAs, NATs, linkRNAs, Hox genes, antisense transcription, Hox cluster evolution

## Abstract

Bilaterian animals operate the clusters of Hox genes through a rich repertoire of diverse mechanisms. In this review, we will summarize and analyze the accumulated data concerning long non-coding RNAs (lncRNAs) that are transcribed from sense (coding) DNA strands of Hox clusters. It was shown that antisense regulatory RNAs control the work of Hox genes in cis and trans, participate in the establishment and maintenance of the epigenetic code of Hox loci, and can even serve as a source of regulatory peptides that switch cellular energetic metabolism. Moreover, these molecules can be considered as a force that consolidates the cluster into a single whole. We will discuss the examples of antisense transcription of Hox genes in well-studied systems (cell cultures, morphogenesis of vertebrates) and bear upon some interesting examples of antisense Hox RNAs in non-model Protostomia.

## 1. Introduction

A little more than a century has already passed since Calvin Bridges, who worked in the laboratory of Thomas Hunt Morgan, revealed the new type of mutations in *Drosophila melanogaster*. Those mutations were localized in bithotax (bx) locus and resulted in a partial transformation of the halter to the wing.

Edward Lewis proceeded with Bridges’ work and within a few decades he described, in detail, the BX-C complex, which controls the morphogenesis of thoracic and abdominal segments of flies [1]. This is how Hox genes were first found, the genes which are universal for all bilaterian animals, control their development and play a huge role in morphological evolution. 

Hox genes certainly are the most studied developmental genes to date. Even now, there are more than 500 papers published every year investigating or discussing their functions. This undying interest in the subject can be explained by the multiple roles of Hox genes in development. These genes work through the whole embryogenesis process starting from the earliest steps of development [2] and until the extreme old age of multicellular animals; the vertebrates which use Hox genes to control homeostasis, being the perfect example [3,4,5,6]. The same genes specify the organization of the body plan of the animals from the largest clade—Nephrozoa. At the same time, they can be easily co-opted into the developmental programs of evolutionarily new structures, such as the photophore of fireflies [7] and hair follicles of mammals [3].

We suggest that the pervasiveness of Hox genes is caused by:Their fundamental role in the ground plan formation (this excludes the loss of Hox genes in most bilaterian animals);The simplicity of DNA-consensus for the binding of Hox homeodomain and the ability of Hox genes to form the dimers with cofactors to make this consensus more complicated;The complex regulation of their transcription.

Figure 1 summarizes the schematic presentations of the known mechanisms of Hox genes’ transcriptional control. This complicated picture raises a number of issues. For example, it is unclear at which evolutionary step this regulatory complexity appeared. What could it be used for if it originates from the common ancestor of Nephrozoa? Which regulatory mechanisms were inherited by modern Bilateria from their common ancestor and which arose independently in different taxons? Which mechanisms are predetermined by the structure of Hox cluster? Are there any mechanisms that can underlie the clusteral structure of Hox genes? To solve these fundamental questions, the several extant monographs do not seem to be sufficient, but it is possible to come closer to an adequate understanding through analysis of separate regulatory mechanisms. In this review, we intend to discuss the accumulated data concerning long non-coding RNAs which are transcribed from the sequences of Hox clusters in the opposite direction, i.e., from the sense (coding) strand. We have focused on these regulatory transcripts because there are more of them in Hox clusters than sense lncRNAs and because they are better studied. In addition, the regulatory potential of antisense molecules is higher due to additional options, such as the formation of duplexes with mRNAs of Hox genes.

Large arrows indicate Retinoic acid pathway (RA), Sonic Hedgehog pathway (SHH), WNT and FGF pathways.

The world of regulatory RNAs is huge. Since the accumulation of data on the structure and function of regulatory RNAs is actively proceeding, even their classification is not yet stable. Traditionally, non-coding RNAs are divided into short (<200 bp) and long (>200 bp) classes [8,9]. Long non-coding RNAs are the largest and the most heterogenic class of ncRNAs. These molecules were found in all the organisms studied to date including viruses. With a few exceptions, they are synthesized by RNA-polymerase II, possess the 5′-cap, polyadenilated and can exist simultaneously in both poly-A+ and poly-A- forms [10]. Their most intriguing characteristic is that in the row of multicellular animals and plants the percentage of lncRNA coding sequences in genome grows together with the increasing morphological complexity, i.e., the increasing number of cell types. For example, the sponge *Amphimedon queenslandica* (Demospongia) has no more than 15 cell types. The transcriptomic analysis of the several developmental stages of *Amphimedon* revealed 2935 lncRNAs, which is ~7.5% of protein-coding sequences (around 40,000 mRNA) of this sponge [11,12]. There slightly more than 400 cell types in the human body, around one third of which are comprised by the derivatives of the neural crest [13]. According to the various databases, the number of genes coding lncRNAs in humans varies from 140,356 (LncBook: https://bigd.big.ac.cn/lncbook/index) to 56,946 (LNCipedia: https://lncipedia.org), which, by a significant extent, exceeds the number of mRNA coding genes (20,352 according to CHESS: http://ccb.jhu.edu/chess accessed on 15 June 2021). 

It is argued that around one third of all the transcripts produced by an eukaryotic cell represent the transcriptional noise, caused by the imperfection of splicing mechanisms and mistakes in the initiation of transcription [14]. One can suggest that a huge number of lncRNA which can be found in metazoan transcriptomes is the result of these mistakes. This seems to be quite a rational view on the issue of huge lncRNA redundancy in comparison with mRNAs.

However, it is recognized that the level of tissue-specificity, i.e., the preferential transcription in the definite cell type, at least of one subclass of lncRNAs—intergenic lncRNAs (lincRNAs)—is much higher than among protein-coding genes [15]. Moreover, the large-scale analysis of transcriptional dynamics of lncRNAs in seven vertebrate species (human, rhesus macaque, mouse, rat, rabbit, opossum and chicken) revealed that during the developmental process there was a transition from the universal and conservative lncRNAs to specific and low conservative ones, which was in good accordance with the expression dynamics of protein-coding developmental genes [16]. 

Antisense lncRNAs are synthesized from the coding DNA strand and can overlap the sequences of protein-coding genes and genes coding other lncRNAs. In the human genome, 44,624 antisense non-coding transcripts were found (according to LNCipedia). It is generally accepted that around 70% of protein coding genes of mammals have antisense transcripts, which are synthesized from their own (independent) or divergent (bidirectional) promotors [17]. Those antisense lncRNAs that are fully or partially overlap the exons of protein-coding genes referred to as Natural Antisense Transcripts (NATs). The first evidence of the existence of the genes coding ncRNAs in Hox cluster appeared in the last century when the transcriptional activity of the bxd-region of *Drospohila* localized 5′ from the *Ubx* gene was analyzed [18]. However, the complicated and variable lncRNA-dependent Hox-regulation is mostly studied in mammals and human cell cultures [19,20,21,22].

## 2. Antisense LncRNAs in Mammalian Hox Clusters

Four mammalian Hox clusters are composed of 39 protein-coding genes, which are transcribed in a spatially and temporally collinear manner in embryonic and definitive tissues excluding the fore- and midbrain. In their 2007 review paper, Denis Duboule classified the clusters of this type as organized [23]. These are small clusters (from 100 to 170 kb) in which all the genes are oriented in a similar fashion and as a result are transcribed in one direction. Hox clusters of mammals are free from any foreign protein-coding genes and almost free from the repeats, but can code miRNAs and lncRNAs. The fact that there are more non-protein coding transcriptional units (TUs) in Hox clusters of humans than of mice [24], and that vertebrates possess, in general, more ncRNAs in Hox clusters than protostomians (though this group is not sufficiently studied) demonstrates the functional importance and the probable contribution of these transcripts to natural selection. 

The first two investigations describing the single lncRNAs of Hox genes were performed in 1995. They demonstrate the transcription of anti-*Hoxd3* [25] and anti-*Hoxa11* (now *Hoxa11os*) in mice [26]. It turned out that these molecules are polyadenylated and have several different isoforms [26]. It was shown that their expression differs from the sense transcripts of sequences they overlap [25,26]. One of the transcripts had distinct nuclear localization [25]. In both cases, antisense RNAs contained the region, which was complementary to the protein-coding part; thus, they belonged to NATs. Surprisingly, anti-*Hoxd3* transcript [25] is still absent in the database, since both possible candidate asRNAs transcribed from this locus—*Hoxd3os1* and *Gm38462*—do not possess exons overlapping homeobox and are expressed in the different parts of the embryo compared with the transcript described in the paper. 

These first described cases looked like anecdotal reports. Only ten years later, the true both-way expression traffic of mammalian Hox clusters was demonstrated by using the available databases (EST and genomic databases) and new genomic technologies (Tiling array, Chip-Seq, CAGE), focused on global transcriptomic analysis [24,27,28,29]. In these large and technically complicated works, the following patterns were revealed: 

First, it turned out that almost the whole of the Hox clusters of humans and mice are transcribed in both the sense and antisense directions [24,27,28]. This transcription was observed in various embryonic and definitive tissues of humans and mice, in the human cell line of teratocarcinoma [24,27] and in fibroblasts [28] from different locations of the human adult body. This last model system revealed 407 discrete transcribed regions in four Hox loci. Only 101 of them refer to the exons of Hox genes [28]. The main part of the transcribed regions (231) are the intergenic areas with three-quarters comprising antisense lncRNAs. These lncRNAs are transcribed from individual or divergent promotors. Second, there are bicystronic and polycystronic transcripts among antisense and sense (including protein-coding) RNAs [29,30]. These molecules are read from extended parts of the Hox clusters (primary transcripts up to 30 Kb), so that they overlap or include the exons of two or more genes.

It is worth noting that not only mammals but also crustaceans possess bicystronic Hox transcripts [31], as well as onychophorans and myriapods [32,33]. Their function is still unclear but it is known that RNAs of this type can be processed to form the normal mRNAs [31], and also the chimeric transcripts that contain the sequence fragments of two neighboring genes [32,33]. The presence of transcripts of this kind can consolidate the Hox cluster. 

Third, antisense lncRNAs of vertebrate Hox clusters are expressed following the rule of spatial collinearity and are often coexpressed with the neighboring protein-coding genes [27,28]. Using the exogenous retinoic acid, the transcription of antisense lncRNAs was induced in the cells of embryonic teratocarcinoma (NT2D1) alongside mRNA transcription. In some cases, lncRNAs were activated even earlier than neighboring Hox genes and could serve as mediators for this induction [27]. Since the initiation of their activation in the experiment was followed by the liberation of chromatin from repression complexes, we can assume that one of the functions of persisting antisense lncRNA expression lies in the prevention of transcriptional silencing [24,27]. It is important to remember that a part of antisense lncRNAs does not co-express with the neighboring genes [28]. Moreover, RT-PCR or tiling array methods do not reveal the spatial transcription pattern in the organ or its anlage. For example, the complementary expression of Hoxa11 and Hoxa11os, visualized in the limb anlage by WMISH [26], would be revealed as co-expression by the aforementioned methods.

Fourth, lncRNAs of Hox genes possess a certain evolutionary conservatism at the level of primary sequences. They are less conservative than protein-coding RNAs, but more conservative to various degrees than non-transcribed parts of the clusters [28]. The homologs of HOXB-AS3 are found in different vertebrates from the elephant shark to humans, but more importantly, there is a homology between the promotors of HOXA-AS3 and HOXB-AS3. These genes can probably be considered as ohnologs, as the protein-coding Hox genes from different clusters. This means that in the single ancestral cluster of the vertebrates, the sequence of this lncRNA already existed [34]. Moreover, a significant number of lncRNAs of Hox genes possess the special type of evolutionary conservatism—syntenic conservatism [24,29,35,36]. In this case, the very position of lncRNAs and their functions are conserved but not their primary sequence. In terms of the evolutionary distance between *Branchiostoma lanceolatum* and *Homo sapiens*, 16 syntenic homologs were revealed [36]. The question about equivalency of these molecules’ function is still open and each case needs individual analysis due to the specificity of Hox traffic regulation. We will discuss one of the examples below. 

The fact that the new class of molecules contained in Hox clusters is functionally significant was indirectly confirmed by the presence of small motifs (6–8 nt), which were specific for the different molecules from spatially variable parts of the body. In other words, lncRNAs from the anterior parts of the body differ by these motifs from the transcripts of posterior parts. Motifs that were specific for proximal and distal body parts were also found [28]. Moreover, Hox clusters of vertebrates do not practically contain repeats, and the small blocks of SINE, LINE and LTR can be only found in the loci where neither sense nor antisense transcription occur [29]. However, for the full study of the functional role of lncRNAs in the processes of development and growth, experimental works on representational models are needed. It was the case that the majority of data concerning Hox lncRNA functions were obtained on a limited number of models, mainly human cell cultures. Table 1 and Table 2 contain the 18 most studied antisense lncRNAs from the four clusters of human Hox genes. One can see that each molecule possesses multiple functions, most of which were described for pathological processes. In most cases, these lncRNAs perform transcriptional silencing, the induction of transcription or its modulation through recruiting the proteins of chromatin remodeling and transcriptional factors to the target sites. Here, lncRNAs work as scaffolds for the assembling of the protein complexes, both repressing and activating. Moreover, most of them work as competing endogenous RNAs (ceRNAs), which scavenge many miRNAs of different types as a sponge. Almost all of these molecules influence Hox gene expression in cis or trans. Hox genes themselves also regulate the transcription lncRNAs. For example, it was shown that HOXB13 directly interact with the promotor and induce the transcription of HOXC-AS3, which further provoke proliferation, migration and invasion of glioblastoma (GBM) cells [37]. 

LncRNAs of Hox clusters are transcribed directly (sense) and in an opposite way (antisense). Though the number of antisense transcripts is considerably larger, there is no impression that they function according to different rules. In fact, the transcripts of both types may overlap the large parts of Hox clusters, may be intronic and intergenic, may work in cis- and trans-mode, possess conservatism to a greater or lesser degree and effectively function as scaffolds. However, the important regulatory mechanisms exist, which can be only performed through the antisense transcripts. In Table 1, the majority of lncRNAs are marked as “linc or NAT” because, among the registered isoforms of one RNA, both types were found. The large part of these molecules indeed have exons that partially overlap with the exons of Hox genes. Theoretically, this means that NATs can interact with mRNAs of Hox genes and either stabilize or cleave them to siRNA, thus participating in post-transcriptional silencing. Moreover, the antisense lncRNAs can repress the transcription of sense transcripts through the mechanism of interference at the level of transcriptional complexes [38]. This mechanism is realized in the cell nucleus. 

**Table 2 jdb-09-00028-t002:** Antisense lncRNAs from four human Hox clusters and their main functions and targets.

As ncRNA	Functions	Mechanism of Work	Localization	Targets	Orthologs	Discovered in	Refs
HOTAIRM1	Control of the cell cycle in the myeloid cell lineage; control of the differentiation of granulocytes; control of neuronal differentiation timing; control of osteogenesis in dental follicle stem cells	Serve as protein scaffolds; Enhancer; Sponges big set miRNAs	Nucleus Cytoplasm	HOXA cluster; NEUROGENIN 2; miR-196b; miR-125b	Chordata	2009	[36,39,40,41,42,43,44]
HOXA-AS2	Promotion of proliferation, migration and invasion in many types of tumors; regulation EMT; negative regulates endothelium inflammation	Serve as protein scaffolds; Sponges big set miRNAs	Nucleus Cytoplasm	c-MYC; EGFR; Bax/TRAIL; EZH2/LSD1; PBX3; NF-kB; miR-373	Primates	2013	[45,46,47,48,49]
HOXA-AS3	Control of cell cycle, proliferation, migration and apoptosis in many types of cancer cells; positive regulation of endothelium inflammation; activation the MEK/ERK Signaling Pathway	Stabilization of HOXA6 mRNA, sponges miR-29c and mir-455-5p	Nucleus Cytoplasm	HOXA6; NF-kB; miR-29c	Homo sapiens	2017	[50,51,52,53,54]
HOXA10-AS	Cell cycle and apoptosis control in glioma, lung adenocarcinoma (LAD), oral cancer and acute myeloid leukemia (AML) cells	?	Cytoplasm	HOXA10; Wnt pathway; NF-kB	Birds Mammals	2018	[55,56,57,58]
HOXA11-AS	Control of the menstrual cycle; cell cycle, proliferation, migration and apoptosis control in many types of cancer cells	Serve as protein scaffolds; Sponges big set miRNAs	Nucleus Cytoplasm	HOXA11; TGF-b pathway, LATS1; CyclinD1; CyclinE; CDK4; CDK2	Rodents Primates Bamboo shark	2002	[59]
HOTTIP	Control of 5′HOXA genes’ transcription during development. Participation in pathogenesis of almost all types of cancer.	Activates HOXA genes through recruiting of WDR5 и MLL	Nucleus	HOXA7-HOXA13; LSD1; EZH2; IL-6; miR-30b	Rodents Primates Bamboo shark	2011	[60,61]
HOXB-AS1	Glioblastoma and endometrial carcinoma and multiple myeloma (MM) promotion	ILF3-mediated activation of HOXB3 and HOXB3 transcription; stabilization of their mRNAs; stabilization of FUT4 mRNA	Nucleus Cytoplasm	HOXB2; HOXB3; Wnt pathway; FUT4; miR-186-5p; miR-149-3p	Homo sapiens	2019	[62,63,64]
HOXB-AS2	Potentially participate in the development of atrial fibrillation	?	?	?	Homo sapiens	2020	[65]
HOXB-AS3	Control of energetic metabolism in the cell through alternating the isoforms of pyruvate kinase M (PKM); promoting of the cancer processes through repression of p53 transcription; activation of PI3/AKT pathway	Codes the conservative peptide of 53 amino acids long, which is important for PKM splicing; Sponges miRNAs	Nucleus Cytoplasm	PKM; DNMT1; p53; I3K-AKT-mTOR pathway; miR-378a-3p	Homo sapiens Rodents	2017	[66,67,68,69,70]
HOXB-AS4	The sequence is differentially methylated in normal and pancreatic cancer cells	?	?	?	Homo sapiens	2018	[71]
HOXB-AS5 or PRAC2	Associated with breast cancer (lncRNA) and protstate cancer (protein)	Encodes 140 aa nuclear protein	Nucleus (protein)	I3K-AKT-mTOR pathway (lncRNA)	Artiodactyla Bats Colugo Primates	2003 (protein) 2017 (lncRNA)	[72,73]
HOXC-AS1	Cholesterol homeostasis participation, inhibition of atherosclerosis; promotion of growth and metastatic formation in several types of malignant tumors	Promotes the transcription and translation of HOXC6; boosts c-MYC mRNA	Nucleus Cytoplasm	HOXC6; miR-590-3p; c-MYC; Wnt pathway; miR-590-3p	Homo sapiens	2016	[74,75,76,77,78]
HOXC-AS2	Promotion of growth and metastatic formation in several types of malignant tumors	Promote HOXC13 transcription; can sponge miR-876-5p to affect ZEB1 expression	Nucleus Cytoplasm	HOXC13; ZEB1; miR-876-5p	Homo sapiens	2019	[79,80,81,82]
HOXC-AS3	Functions under the direct control of HOXB1. Promotion of growth and metastatic formation in several types of malignant tumors	Promote the transcription of 5’HOXC genes; stabilizes HOXC10 mRNA; can sponge miR-3922-5p, impairs the maturation of miR-96	Nucleus	HOXC8; HOXC9; HOXC10; HOXC11; HOXC12; HOXC13; YBX1; thymidine kinase 1 (TK1); FOXM1; miR-96; miR-3922-5p	Homo sapiens	2018	[37,81,82,83,84,85,86,87,88,89]
HOTAIR	Reprogramming of chromatin state to promote cancer metastasis; PRC2 and PRC2-independent induction of transcriptional repression. Promotion of growth and metastatic formation in several types of malignant tumors	Scaffold: A 5′ domain of HOTAIR binds PRC2, whereas a 3′ domain of HOTAIR binds the LSD1/CoREST/REST complex; Sponging big set microRNA	Nucleus Cytoplasm	HOXD cluster (40 Kb in 5′area) HOXA1; HOXA5; HOXC11; p53; p27; E-cadherin; NOTCH1/JAGGED1; SNAIL; GLI2; Protocadherin 10; Wnt pathway; Dozens of miRNAs, critical for proliferation and differentiation control	Rodents Carnivores Primates Marsupials	2007	[28,35,90,91,92,93,94,95,96,97,98,99,100,101,102,103,104,105,106,107,108,109,110,111,112,113,114,115,116,117,118,119]
HOXC13-AS	Promotion of proliferation, migration and invasion of cells of several cancer types	Sponging big set microRNA	Cytoplasm	HOXC13; c-MYC; miR-383-3p; miR-497-5p	Homo sapiens	2019	[120,121,122,123,124]
HAGLR or HOXD-AS1	Impairment of HAGLR regulation (up or down) to promote growth and metastatic formation of several types of malignant tumors	Binds to WDR5 and EZH2 for activation and repression of target genes	Nucleus Cytoplasm	HOXD3; JAK2/STAT3 pathway; Wnt pathway; Ras/ERK pathway; TGF-β pathway; p57; miR-133a-3p; miR-133b; miR-130a; miR-185-5p	Rodents Primates	2014	[125,126,127,128,129,130,131,132,133,134,135,136,137,138]
HOXD-AS2	Downregulation of HOXD-AS2 significantly promotes the progression of gastric cancer	?	Cytoplasm	HOXD8; PI3K/Akt pathway	Homo sapiens	2018	[139,140,141,142,143,144,145,146,147,148,149]

Surprisingly, not all lncRNAs can be positively considered as non-protein coding. The small (less than 100 codons in length) open reading frames (small open reading frames; smORFs) are found in many lncRNAs. Their functional coding potential was under debate for a long time, but the peptides were later found in mammals; these were synthesized from “non-coding” templates. Around a dozen and a half short molecules, from 9 to 250 amino acids long, are synthesized from the templates of precursors of miRNAs, lncRNAs and even from one transcribed repeat [150,151]. The functional meaning of some of the peptides from lncRNAs of Hox clusters was shown in experiments. HOXB-AS3 lncRNA codes a peptide that is 53 amino acids long. It turned out that this peptide functions as a switch in the energetic metabolism of the cell, since it controls the spicing of one of the most important catabolic enzyme—pyruvate kinase M (PKM). In the presence of this peptide, the isoform M1 of kinase is synthesized, which shifts the energetic metabolism towards the oxidative phosphorylation. In the absence of the peptide, another isoform of the pyruvate kinase M—M2—prevails in the cell, which promotes the glycolysis. The first isoform dominates in the mst differentiating somatic cells, while the second one is needed in the actively proliferating stem cells that use anaerobic glycolysis as a source of “fast” ATP synthesis. The majority of human tumors demonstrate the reduced transcription of HOXB-AS3 and start to synthesize PKM2. The metabolism of the tumor cell is forced to undergo glycolysis since the mitochondria, the membranes of which are used for oxidative phosphorylation, are not functional [66]. Importantly, Hox genes do not function in the actively proliferating ESM, and the synthesis of HOXB-AS3 alongside the regulatory peptide occurs later, when the intensity of embryonic cell proliferation goes down. 

We would like to discuss the high evolutionary and regulatory plasticity of Hox lncRNAs using the examples of the two most studied molecules. In 2007, the first human antisense lincRNA—HOTAIR—was discovered and functionally characterized [28]. It is transcribed from the region between HOXC11 and HOXC12 and functions as a scaffold for assembling of the protein complexes PRC2 (Polycomb Repressive Complex 2) and LSD1 (Lysine-Specific Demethylase 1). Due to these two complexes, HOTAIR realizes two functions at the same time—the methylation of H3K27 and demethylation of H3K4me2 [90]. HOTAIR starts to work early in human embryogenesis and is needed for transcriptional silencing of the 5′-area of HOXD cluster [28]. Moreover, it controls the work of a large group of miRNAs which are critical for development and differentiation. Dozens of transcriptional factors and components of signaling pathways function under the direct or indirect control of HOTAIR. The defects in regulation of HOTAIR transcription were found in many types of malignant tumors. Surprisingly, this functional load does not bring about evolutionary conservatism. The similarity between human and mouse HOTAIR comprises only 55% [91]. Besides, TUs, which are positionally equivalent to HOTAIR, are transcribed from the homological cluster regions of many mammals including kangaroos [92]. Contrary to the human homolog, murine Hotair does not contain the motifs for interaction with the proteins of chromatin remodeling and cannot participate in cis- or trans-silencing, which was demonstrated by the deletion of a large fragment of the HoxC locus [93]. The absence of Hotair does not critically interfere with the development of mice. The mutational effects did not exceed the natural variability. Therefore, Denis Duboule et al. have termed the effects caused by Hotair mutations “homeopathic rather than homeotic” [94]. However, if the deletion of the certain nucleotides of Hotair is performed, the impairment of the transcription of neighboring protein-coding Hox genes occurs also due to the formation of new lncRNAs—“ghostair” and “antiHotair”—which form as a result of this deletion. Consequently, the punctual deletion of Hotair in mice can provoke the formation of a homeotic phenotype that is close to the expected one, taking into account the conservative function of human and murine Hox orthologues [20,94]. Thus, the functional conservatism of lncRNAs cannot be estimated through the mutant phenotype without detailed analysis of mechanisms underlying this phenotype. 

Another fascinating example of the functional plasticity of lncRNA in one organism is an antisense lincRNA HOTAIRM1 (HOXA Transcript Antisense RNA, Myeloid-Specific 1). The HOTAIRM1 gene is localized between HOXA1 and HOXA2. This lncRNA is conservative at the level of its position in the cluster. Its syntenic homologs are found in birds, amphibians, bony fishes and even in lancelet [36]. The global studies on the role of Hotairm1 in the mammalian development were not yet performed but it was shown that it controls the differentiation of myelopoietic cells and osteogenesis in the descendants of dental follicle stem cells [19,39]. In the cell line NT2-D1 (embryonal pluripotent carcinoma), Hox gene transcription can be induced by retinoic acid (RA). In this cell line, HOXA1 and HOXA2 are located topologically close to HOXA4, HOXA5 and HOXA6 due to chromatin looping before the RA induction. After the induction with RA, the transcription of all Hox genes is initiated, but HOTAIRM1 modulates their work in such a manner that they all become collinear at the transcriptional level [152]. 

The synthesis of two different isoforms of HOTAIRM1 containing three exons—the long non-spliced and the short spliced ones—is also initiated by RA. The first isoform recruits the protein complex UTX/MLL (H3K27-demethylase/H3K4 methyltransferase) that initiates euchromatization of HOXA1 and HOXA2 loci and, thus, their transcription. The short HOTAIRM1 isoform interacts with PRC2 and lowers the transcription level of HOXA4, HOXA5 and HOXA6. This process is followed by reassociation of the proximal and distal parts of the cluster, but the initial spatial vicinity between HOTAIRM1 gene and other proximal genes allows the short repressing isoforms to reach the target sites by diffusion. This truly concise and elegant mechanism is not realized in another cell line NB4 (acute promyelocytic leukemia) with a different chromatin landscape. HOTAIRM1 isoform from NB4 cells contains two but not three exons and does not control the proximal genes [152]. It was recently discovered that HOTAIRM1 represses the transcription of NEUROGENIN 2 in Hox-independent manner and, thus, is localized at the top of regulatory cascade, which regulates the timing of neuronal differentiation [40]. 

Thus, we can safely assume that lncRNAs of mammalian Hox clusters are numerous, multifunctional and changeable (Table 2). At the same time, they can be observed in the same regulatory continuum with conservative Hox genes and both participate in the reciprocal regulation [24,28,37,152]. In this connection, taking into account the large set of functions that are critical for tissue homeostasis (and probably for development), it is still an enigma as to how lncRNAs can demonstrate such a high evolutionary plasticity. The most provocative aspect here is the structural and functional divergence of the human and murine HOTAIR. 

We suggest that this evolutionary paradox can be solved if observed from the ontogenetic point of view. The multiple and variable functions of lncRNAs may be easily delegated. They can be duplicated in the organism by other components of GRNs and, in case of their dysregulation, this impairment can be compensated during the developmental process. Local GRNs that control the differentiation of multi-, bi and unipotential stem cells of the adult organism do not possess the compensatory potential of the complicated multilevel embryonic GRNs. This can explain the participation of Hox lncRNAs and Hox genes themselves in the tumor formation. 

Thus, in the hyperconservative compact mammalian Hox clusters, the molecular machinery of antisense ncRNAs is internalized, constantly generating changes in their regulation. The range of this variation is defined by the stabilizing selection that happens also at the level of the cell populations in the developing embryo. This is a breeding ground for ontogenetic and evolutionary variability. 

## 3. Antisense LncRNAs in Hox Clusters of Protostomian Animals

The increasing interest in lncRNAs raised the number of studies in which different insects, including those that did not belong to the classical models of molecular and genetic research, were used [50,83,153,154,155,156,157]. From these studies, it has emerged that insects possess lesser amounts than vertebrates but still significant repertoires of these molecules (from 2949 for *Anopheles gambiae* to 11,810 for *Bombyx mori*).

There is no surprise that Drosophila is the best studied protostomian animal in the sense of genome structure and molecular architecture of development. It is worth noting that the first genes that were mapped in the BX-C complex in the classical paper of E. Lewis in 1978 [1] belonged mostly not to the protein-coding sequences but to the regulatory elements that, nevertheless, were transcribed [18,139]. 

In a research study performed in 2002, a series of 1–2 Kb Dig-probes was synthesized. These probes overlapped the intergenic region between abd-A and abd-B genes of *Drosophila*. This intergenic region of 100 kb length was called iab (infraabdominal) and contained cis-regulatory elements that control the nearby Hox genes [139]. Almost all the probes used revealed the transcription of unknown LncRNAs in fly embryos. This transcription preceded Hox mRNA expression and demonstrated spatial collinearity that was similar to mRNA patterns. 

Only three probes revealed antisense transcription and, at present, we can identify them with two ncRNAs presented in FlyBase (http://flybase.org): Dmel\lncRNA:iab4 (CR31271) and Dmel\lncRNA:CR43617. The first transcript belongs to the specific type of lncRNAs that are processed to become miRNA (mir-iab4). Mature mir-iab4 takes the *Ubx* gene under negative control [158,159]. The biological activity of lncRNA iab-4 seems to be limited by this function. The mir-iab4 sequence is evolutionarily conserved and can be traced in insects, chelicerates and crustaceans [160]. The function of the second transcript remains unclear. 

In the modern database FlyBase (http://flybase.org) dozens of lncRNAs are presented, including those which are produced from ANTP-C and BX-C regions, but their functions are still unknown. It is worth mentioning that there are no NATs among these non-coding transcripts, but intronic and intergenic antisense lncRNAs are highly presented. Since the Hox cluster of *D. melanogaster* as well as of other drosophilids is disrupted and contains microinversions, it cannot be regarded as an ancestral cluster of insects [161]. One can assume that the repertoire of Hox-associated lncRNAs is depleted in the fly but is preserved in other species, for example, in the intact Hox cluster of the beetle *Tribolium castaneum*.

The work of Shippy et al. (2008) describes the transcriptional activity of the beetle Hox cluster studied by tiling arrays [162]. It turned out that the Hox cluster of *T. Castaneum* produced multiple non-coding transcripts between 0 and 72 developmental hours. However, the authors described only two antisense LncRNAs, one of which is transcribed from the intergenic region between Antp and Ubx genes, and the second—from the intron of Ubx gene. Thus, despite the huge amount of data, the NATs of Hox genes of insects either do not exist or they are not yet found. 

Two research studies were focused on lncRNAs in crustaceans; notably, in both cases, single-molecule real-time (SMRT) sequencing by PacBio was used. A total of 3958 lncRNAs were found in pacific white shrimp (*Litopenaeus vannamei*) [163]. This fact demonstrates the essential similarity of hexapods and crustaceans in the number of lncRNAs. Wan et al. (2019) managed to sequence 23,644 long non-coding RNAs in crab *Scylla paramamosain* (Decapoda). However, at the time of the paper’s publication, the genome assembling was not yet finished. The authors admitted that they were not able to assess the false rate of identified lncRNAs without the fully assembled genomic data [164]. 

One gets the impression that despite a significant number of the assembled genomes in the Pancrustacea clade being of high quality, there is still the lack of data about the functions and even the number of lncRNAs. It is worth noting that it is the representatives of crustaceans who possess the most compact Hox clusters among all arthropods—192.8 Kb in *Paracyclopina nana* (Copepod) and 324.6 Kb in *Daphnia magna* (Branchiopoda) [165,166]. Here, as well as in the case of vertebrates, the mechanisms consolidating the cluster exist but are not yet studied. 

Myriapoda is a sister group to Pancrustacea. The genomes of the following three myriapods are assembled to date: *Strigamia maritima*, *Helicorthomorpha holstii* and *Trigoniulus corallines*. These belong to the two large sister branches—Centipede and Millipede [167,168]. The authors did not focus on the analysis of lncRNAs but clarified the genomic organization of myriapod Hox clusters and revealed their important ancestral feature. It turned out that Hox clusters of *Strigamia* (Centipede) and *Trigoniulus* (Millipede) are flanked with the orthologs of *Eve-skipped* (*eve*) gene on their 5′-ends. The traces of this ancient synteny were earlier found in chordates and some cnidarian species. This feature, together with the compactness of Hox clusters (457 Kb from *labial* to *eve* in *S. maritime*), make the myriapods a prospective model for studying the mechanisms of Hox-regulation, including those that involve NATs. Myriapods were the first Protostomia for which Hox-associated NATs were found and then studied [32,33]. NAT of *Ubx* (*aUbx*) of *Strigamia* is cloned (GenBank: DQ368689.1), occupies the region of 1051 nt and is complementary to 3′UTR of *Ubx* mRNA (GenBank: DQ368688.1) within sequences that are at least 226 bases long. *aUbx* promotor is localized 3′ from the sequence of *Ubx* gene and closer to 5′-end of *Antp* gene. *aUbx* starts to express earlier and more anterior than its sense transcript, simultaneously with *Antp* mRNA. In addition, sense and antisense *Ubx* patterns look mutually exclusive [32]. 

In two different myriapod species—the millipede *Glomeris marginata* and the centipede *Lithobius forficatus*—*aUbx* RNAs were revealed with sense probes. The expression patterns of antisense RNAs were complementary to *Ubx* mRNAs, which indicates the high conservation of this regulatory relationship in myriapod clade [33]. Outside myriapods (in onychophora *Euperipatoides* and spider *Cupiennius*), no *aUbx* or any other NATs transcribed from Hox loci were found, at least by sense probes [33]. 

Among spiralian animals—the second protostomian clade—antisense lncRNAs of Hox genes were found in nereidid annelids [169,170]. Using sense probes, our working group showed the presence of NATs of *Hox5* and *Hox7* in the postlarval development and regeneration of *Alitta virens*. Later, we revealed the antisense transcription for almost all genes of *A. virens* and *Platynereis dumerilii* Hox clusters [171]. We managed to clone some of these NATs. The structure of the *P. dumerilii* Hox cluster is not yet published. The *A.virens* Hox cluster is studied by pulsed-field gel electrophoresis [172]. According to these data, the cluster is not atomized and does not exceed 2.4 Mb in length. In general, we observe a rich repertoire of antisense patterns for nereidid Hox genes, some of which are complementary to mRNA patterns and some demonstrate the large overlapping zones between sense and antisense areas of transcription. NATs of Hox genes are revealed in the developing segmented larva, in growing and regenerating juvenile worms and, presumably, are present at all stages of the worm’s life cycle. The functional role of these transcripts is unclear and needs to be studied in further experiments. From the analysis of the expression of *Avi-antiHox5*, it is obvious that the antisense transcript is quickly up-regulated at the amputation site while *Hox5* mRNA gradually vanishes from this area [170]. 

In general, despite the huge amount of genomic data and detailed analysis of whole genome transcriptional dynamics performed for model and non-model protostomian animals, little attention is focused on regulatory RNAs, particularly NATs. The only hope is that “big data” for the most transcriptomes studied by the modern methods already contain the needed information, which should be included in the analysis. 

## 4. The Implications for the Uprise of Antisense LncRNAs in Hox Clusters and the Reasons for Their Evolutionary Maintenance

Thus, Hox-associated antisense lncRNAs were found in mammals, insects, myriapods and nereidid polychaetes (Figure 2). This regulatory principle may have already existed in the common ancestor of Nephrozoa. Nevertheless, it does not appear that those molecules were found in all model systems where they were looked for. There are probably some specific “demands” of natural selection on the functioning of the Hox cluster which support—or, on the contrary, alleviate—the priority of this regulative mode. 

We can assume that one of the reasons for the large number of Hox-associated antisense lncRNAs (as well as sense ones) in mammals and other vertebrates is the necessity for dose compensation of Hox-ohnologs. This compensation can be realized due to the delicate epigenetic tuning, where the mediators between Hox sequences and remodeling proteins or transcriptional factors are lncRNAs working in both cis- and trans-positions. However, this function cannot be primary and the only one since, among the animals possessing one Hox cluster, the profound antisense transcription was also unraveled (myriapods, annelids). 

The complicated regulation of gene functioning demands the extensional regulatory sequences, but it is not true for mammals. There are probably restrictions of the Hox cluster size caused by the specificity of their early activation in vertebrate development. During gastrulation, the consecutive liberation of Hox loci from heterochromatized territories occur from 3′ to 5′ ends of the clusters and, thus, the sequential temporal activation of Hox genes is manifested [173]. This process is directly connected to the speed of primary mesenchymal cells’ (PMCs) ingression through the primary stick [174]. The more 5′-Hox genes are activated in the ingressing cells, the slower they are internalized and the more posterior is the location they obtain in the end. By these means, the spatial Hox code of vertebrates is realized through the temporal dynamics of euchromatization, which, in turn, depends on the physical size of Hox locus. An abundance of intergenic and polycistronic lncRNAs (sense and antisense) in Hox clusters may function as mediators in the process of early euchromatization and stabilize the local epigenetic states. They can also keep the cluster from relaxation and disruptions. 

It is unknown whether the compact cluster of vertebrate ancestors allowed them to use the system of lncRNAs, or whether these molecules themselves possessed sufficient functional importance to resist the natural selection pressure and keep the Hox cluster from relaxation and reorganization. To establish the cause and effect in this scenario, the lncRNAs’ loss-of-function experiments in vertebrates and comparative data concerning Hox traffic in the species with variably organized clusters are needed. At this point, it is worth mentioning that animals with disorganized clusters that include a gap (for example, Drosophila) possess very poor repertoires of lncRNAs in comparison with vertebrates. In the lancelet, which preserved the integral cluster, the syntenic homolog of Hotairm1 was found even though it is absent in tunicates and appendicularians with disorganized and atomized clusters [36]. Interestingly, Hotairm1 was not found even in cyclostomes with an integral but relaxed Hox cluster [36,175]. 

Recent studies have revealed a fascinating detail. It turned out that antisense transcription is not an exotic mode of regulation but a natural means of maintaining the chromatin of functional genes in a dynamic state. By the dynamic state, the authors mean the high level of histone turnover, i.e., the circulation of histones in the promoter and gene body [176]. It was shown in yeast studies (S. cerevisiae) that antisense transcription produces dynamic chromatin [176]. If we imagine that the gene has a final number of configurations at the chromatin level and each of these configurations corresponds to a certain level of expression, than the dynamic chromatin allows the gene to switch between these configurations [176]. This can be the reason why the genes that change their expression level in response to stress or environmental changes most commonly possess the antisense transcripts. Moreover, antisense transcription initiated by bidirectional promotors can spread the regulatory signals from on locus to the neighboring genes [177]. There is one more remarkable detail. The antisense transcription represses poorly activated sense transcription but if the sense transcription is strong enough, the antisense transcription vanishes itself [177]. This means that the sense transcript will only be transcribed in the presence of the threshold amount of the activator. This, in turn, predetermines the differential gene activation in the morphogen gradient. Genes comprised into well-ordered clusters can easily utilize these basic characters for co-regulation. 

From this perspective, we can view the antisense transcription in Hox clusters from a different angle. Mammals and nereidid annelids use Hox genes for their whole life for development, growth, and reparative and physiological regeneration. This means that they have a constant demand for dynamic chromatin in Hox loci. Centipede Strigamia forms a finite number of segments in embryogenesis (epimorphic type of development). However, the basal myriapods form the terminal number of segments only after a few moults have passed (hemianamorphic type of development). Thus, they also need the postembryonic work of Hox genes [178]. One can probably expect that antisense Hox lncRNAs will be found in those animals that retain the integral clusters and keep using Hox genes in postembryonic life. 

## 5. Conclusions

The discovery of regulatory RNAs, most particularly lncRNAs, changes our understanding of metazoan genome functioning. Hox clusters of bilaterian animals are among the most studied loci. So far, with the detection of complicated bidirectional transcriptional traffic in Hox clusters, we can feel ourselves to be in a position akin to that of a person who discovered that the decorative ornaments in the margins of a long familiar book are not just ornaments but are complete texts, with different grammatical and syntactical features, which contain the key to a deeper understanding of the main text. All antisense lncRNAs in human Hox clusters are therapeutic targets for malignant tumors, and their careful study has profound practical meaning. However, that is not the only point. We assume that the understanding of Hox clusters’ evolution cannot be full without consideration of the regulatory transcripts encoded in these clusters. 

## Figures and Tables

**Figure 1 jdb-09-00028-f001:**
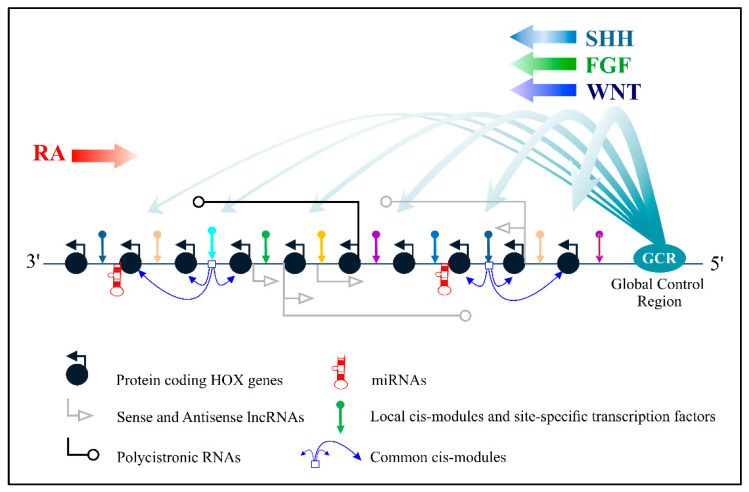
The main pathways for controlling the transcription of Hox genes.

**Figure 2 jdb-09-00028-f002:**
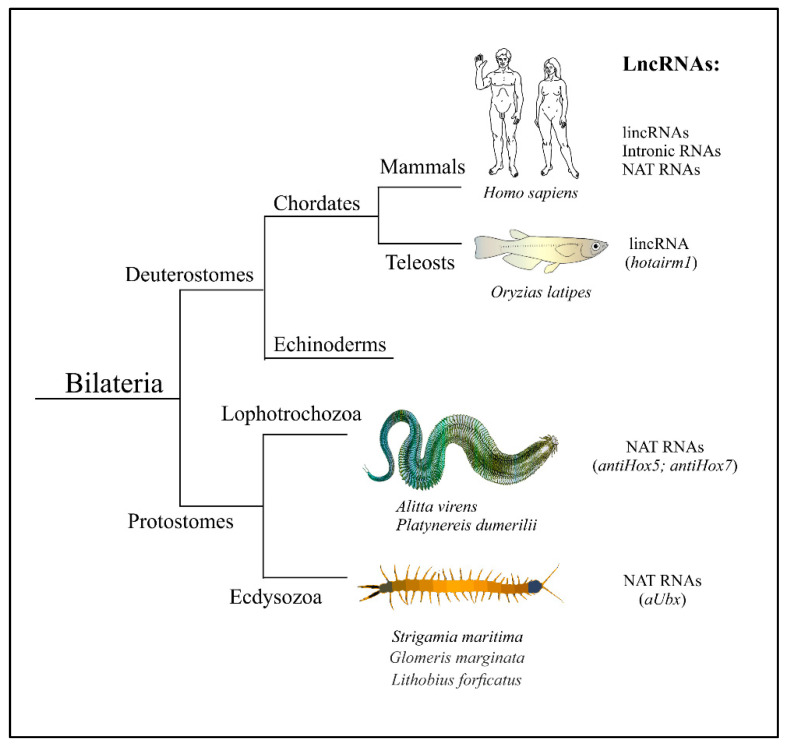
Hox-associated lncRNAs are found in bilateral animals belonging to three main evolutionary lineages—Deuterostomia, Lophotrochozoa, and Ecdysozoa. Individual human lncRNAs are listed in Table 1.

**Table 1 jdb-09-00028-t001:** Antisense lncRNAs from four human Hox clusters. The zones of the overlap of lncRNA exons and RNA exons from the opposite (template) strand are shown in red.

LncRNA	Length (nt)	Type of LncRNA	Position in the Hox Cluster
**HOXA Cluster**			
*HOTAIRM1*	4000; 1052; 783	Linc	*HOXA1→HOXA2*
*HOXA-AS2*	1048	Linc or NAT	*HOXA3→HOXA4*
*HOXA-AS3*	3918; 3992	Linc or NAT	*HOXA4→A5,→A6→HOXA7*
*HOXA10-AS*	1161	Linc or NAT	*HOXA9→HOXA10*
*HOXA11-AS*	1628	Linc or NAT	*HOXA11→HOXA13*
*HOTTIP*	4665	Linc or NAT	*HOXA13→EVX1*
**HOXB Cluster**			
*HOXB-AS1*	797	Linc or NAT	*HOXB2→HOXB3*
*HOXB-AS2*	3594	NAT RNA	*HOXB3*
*HOXB-AS3*	785; 611; 549; 545; 514; 452; 446; 336	Linc or NAT *	*HOXB4→B5→HOXB6*
*HOXB-AS4*	543; 513	Linc	*HOXB9→HOXA13*
*PRAC2*	1193; 560; 518; 503; 448	Linc *	*HOXB9→HOXA13*
**HOXC Cluster**			
*HOXC-AS1*	548	Intronic	*HOXC9*
*HOXC-AS2*	504	Linc or NAT	*HOXC9→HOXC10*
*HOXC-AS3*	368	Linc or NAT	*HOXC10→HOXC11*
*HOTAIR*	2370; 2364; 2337	Linc	*HOXC11→HOXC12*
*HOXC13-AS*	1408	Linc or NAT	*HOXC9→*
**HOXD Cluster**			
*HAGLR*	4086; 4037; 4007; 3942; 3923; 3905; 3893; 3891; 3821; 3812; 3794; 3782	Linc or NAT	*HOXD1→HOXD3*
*HOXD-AS2*	692	Linc or NAT	*HOXD4→D8→D9→HOXD10*

The asterisk (*) marks the molecules encoding the peptide (HOXB-AS3) and protein (PRAC2).

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
