# Peer review of "There and Back Again: Hox Clusters Use Both DNA Strands"

_jdb, 2021, doi:10.3390/jdb9030028_

Round 1
Reviewer 1 Report
Recently, relations of various kinds of cancer and long non-coding RNAs from Hox gene cluster have been reported. This review for the lncRNAs by E. L. Novikova and M. A. Kulakova is a timely summary looking back for lncRNAs from Hox gene cluster and bring up problems for the lncRNAs. This review is appropriate to Journal of Developmental Biology.
I want to suggest only one point.
I know that the major subject in this review is lncRNAs transcribed from the sense strand. However, is not there any lncRNA transcribed from anti-sense strand of Hox gene cluster? If there are such lncRNAs, the description of lncRNAs from anti-sense strand in Fig. 1 might be helpful to the readers to recognize the complexity of transcription and regulation of the Hox gene cluster region. Is there any difference in expected functions between lncRNAs transcribed from the sense and anti-sense strands? The reason why only lncRNAs from sense strand is the main subject should be mentioned clearly.
Author Response
Dear Reviewer!
We are grateful for your comments. We made the appropriate corrections. In the Introduction, after the phrase: “In this review we would like to discuss the accumulated data concerning long non-coding RNAs which are transcribed from the sequences of Hox clusters in the opposite direction, i.e. from the sense (coding) strand ”, we inserted a clarification:
"We have focused on these regulatory transcripts because there are more of them in Hox clusters than sense lncRNAs and because they are better studied. In addition, the regulatory potential of antisense molecules is higher due to additional options, such as the formation of duplexes with mRNAs of Hox genes."
In addition, we have corrected the Fig.1 by inserting sense lncRNAs into the scheme.
Thank you for your attention to our article. We hope it became better perceived by its future readers.
Best Regards,
Аuthors

Reviewer 2 Report
This is an interesting and timely review of lncRNAs associated with Hox gene clusters. It takes both an evolutionary and a molecular genetic viewpoint. Perhaps most enlightening is the scale to which lncRNAs are produced from Hox clusters. This information is collected and discussed in one place to better help interested parties review this field.
I have only a few points that could be addressed to help make the review more accessible to a more general audience.
The written English is generally excellent, and I am always gratified when non-native speakers write reviews of this quality for English audience. However, there are a few awkward sentences that left me unsure as to the meaning.
For example
"The most intriguing is that in the row of multicellular animals and plants the percentage of lncRNAcoding sequences in genome grows together with the increasing morphological complexity if under the morphological complexity we understand the number of cell types."
I believe the authors are saying that the number of cell types is being used as a proxy for complexity, but I'm not completely sure. This sentence could be rewritten to be more straightforward.
"It is argued that around one third of all the transcripts produced by an eukaryotic cell represents the transcriptional noise, caused by the imperfection of splicing mechanisms and mistakes in the initiation of transcription [14]. This seems to be quite rational opinion on the issue of huge lncRNA redundancy."
I'm unclear as to the the meaning of this, especially the last sentence, how does this relate to lncRNA redundancy?
Otherwise, I would make a few suggestions
On page 4 the authors refer to "very long molecules (up to 30 kb)" please provide information on how those were determined to be one single transcript. Later in the review there is a discussion of long-read sequencing, so please clarify if that was used for the example above.
The absence of Hotair does not critically impair the development of the mouse and minor mutational effects can be called not “homeotic” but “homeopathic”
Please define what is meant by "homeophathic" in this context.
Author Response
Dear Reviewer!
Thank you for your comments. We tried to make the text more readable. We made the first problematic sentence shorter and simplified it:
«The most intriguing is that in the row of multicellular animals and plants the percentage of lncRNA coding sequences in genome grows together with the increasing morphological complexity, i.e. the increasing number of cell types.»
We tried to clarify the next sentence that caused questions.
«It is argued that around one third of all the transcripts produced by an eukaryotic cell represents the transcriptional noise, caused by the imperfection of splicing mechanisms and mistakes in the initiation of transcription [14]. One can suggest that a huge number of lncRNA which can be found in metazoan transcriptomes is the result of these mistakes. This seems to be quite rational view on the issue of huge lncRNA redundancy in comparison with mRNAs.»
The long molecules (up to 30 Kb) mentioned in the paragraph on polycistronic RNAs are primary transcripts. Since we are talking about humans and model mammals, the size of these primary transcripts was determined by aligning the already spliced RNAs on the assembled genomic loci. Of course, we should have clarified this in the article:
«These molecules are read from extended parts of the Hox clusters (primary transcripts up to 30 Kb), so that they overlap or include the exons of two or more genes.»
We write about the "homeopathic" effects of the Hotair mutation by citing a Denis Duboule et al. [46]. The authors of this article mean that the observed developmental abnormalities in Hotair - / - mice do not exceed the natural variability observed in wild-type mice belonging to the strain used.
We'll explain it like this:
«The absence of Hotair does not critically interfere with the development of mice. The mutational effects did not exceed the natural variability. Therefore, Denis Duboule et al. have termed the effects caused by Hotair mutations "homeopathic rather than homeotic" [46].»
Thank you for your attention to our article. We hope it became better perceived by its future readers.
Best Regards,
Аuthors
